# Design and Characterization of Wideband Terahertz Metamaterial Stop-Band Filter

**DOI:** 10.3390/mi13071034

**Published:** 2022-06-29

**Authors:** Hao Li, Junlin Wang, Xin Wang, Yao Feng, Zhanshuo Sun

**Affiliations:** College of Electronic Information Engineering, Inner Mongolia University, Hohhot 010021, China; 32056056@mail.imu.edu.cn (H.L.); 32056112@mail.imu.edu.cn (Y.F.); 32056106@mail.imu.edu.cn (Z.S.)

**Keywords:** terahertz metamaterials, stop-band filter, ultra-wideband, steep cut-off

## Abstract

We propose and design a metamaterial broadband stop-band filter with a steep cut-off in the terahertz region. The filter is based on the flexible structure of metal-dielectric-metal-dielectric-metal (MDMDM). Simulation results show that the filter has a center frequency of 1.08 THz, the square ratio reaches 0.95, and the −20 dB bandwidth reaches 1.07 THz. In addition, it has excellent flat-top characteristics with an average transmission rate in the resistive band of no more than 5%. The relative bandwidth has been up to 99%, and stopband absorption rate has reached more than 98%. The effects of the main structural parameters on the transmission characteristics are discussed. The role of each layer of metal in the filter is explored by studying the effect of the variation of the number of metal layers on the filter. The symmetry of the structure ensures the polarization insensitivity of the filter at normal incidence. The correctness of the simulation results was verified by analyzing the effective permittivity and magnetic permeability. To investigate the transmission characteristics of the metamaterial filter in-depth, we analyzed the electric field strength and surface current distribution at the center frequency of the filter. The designed terahertz filter may have potential applications in terahertz communications, sensors, and emerging terahertz technologies.

## 1. Introduction

Metamaterials are artificial composites, materials with a specific electromagnetic response, which have received increasing research and attention in the last decade. By proper utilization of metamaterials, exotic phenomena such as negative refractive index [1,2], perfect lensing [3], and invisibility [4,5] can be achieved. The unique electromagnetic (EM) response generated by metamaterials is particularly valuable in the terahertz state, where most naturally occurring materials exhibit a weak EM response to terahertz waves. The terahertz region, typically defined between 0.1 and 10 THz, remains the least developed region of the EM spectrum due to the lack of effective terahertz sources, detectors and functional devices, and the advent of metamaterials is expected to narrow the “terahertz gap” [6,7,8].

Terahertz filters are one of the most important terahertz functional devices for terahertz imaging, spectroscopy, and terahertz communications [9,10]. A variety of metamaterial-based filters have been extensively investigated [11,12]. In 2011, N. R. Han et al. reported a broadband frequency response from multilayer metamaterials by superimposing individual two-dimensional metamaterials together to construct a broadband filter with a bandwidth of 0.38 THz and noted that multilayer metamaterials provide a promising approach for narrow-band extension to broadband [13]. In 2012, Li et al. found that multilayer microstructures consisting of loop resonators exhibit the properties of broadband stop filters. By studying the near electric field distribution and using the coupled “Drude–Lorentz-type” resonance model, it was finally concluded that the mechanism of broadband attenuation was attributed to the longitudinal coupling between the layers and the enhanced resonance within the planar microstructure [14]. In 2021, Ruan et al. proposed a broadband metamaterial filter with a steep cut-off in the terahertz region. The filter has a center frequency of 125 GHz and the high squared reaches 0.92. In addition, it has excellent flat-top characteristics with a stopband absorption of more than 95%. This work provides a method for designing wideband stopband filters with high squared ratios with potential applications in terahertz communications and sensors [15].

It is well known that bandwidth and squared ratio are the key performance parameters of filters. Due to the strong electromagnetic resonance effect, metamaterial filters in general have narrow bandwidth and small square ratio [16]. However, trying to improve the square ratio of the filter is challenging for the filter design because the filtering mechanism of metamaterial filters is very complex and the multiple coupling effects make it difficult to improve the steep cut-off characteristics of the filter [17]. In this paper, a wide-stopband terahertz filter with high squared ratio is proposed, and this metamaterial filter is simulated based on the CST2020 simulation software. Its transmission curve is studied and the transmission mechanism of terahertz waves in the filter is analyzed in detail by electric field and surface current distribution. Due to the good performance of the designed filter, it can be applied in the fields of electromagnetic shielding and terahertz communication.

## 2. Design and Simulation

The designed wide-stopband filter in this paper consists of a metal-dielectric-metal-dielectric-metal (MDMDM) five-layer structure as shown in Figure 1. The three metal layers have the same structure, and the metal material is a gold thin film with a conductivity of 4.561 × 107 S·m^−1^. The polyimide is utilized as a dielectric layer with permittivity and loss tangent of ε = 3.5 and tan δ = 0.02, respectively. All parameters of the designed MDMDM structure are set as shown below: cell period P = 125 μm, length of the metal ring L = 110 μm, width of the metal ring w = 8 μm, thickness of the dielectric layer d = 20 μm, and thickness of the metal layer t = 1 μm.

The broadband filter designed in this paper is computer simulated and optimized using the frequency domain solver of CST Microwave Studio. In order to simulate an infinitely periodic cell, periodic boundary conditions are selected along the x-y plane during the simulation, while open boundary conditions are selected along the z plane during the simulation. In addition, the electric (E) and magnetic (H) fields are parallel to the incident plane, while the propagating wave vector (k) is perpendicular to the structure plane. To explore the filtering performance of this filter, the transmission coefficient of the normal incident wave is focused on. The transmission curves are obtained by simulation as shown in Figure 2. The simulation results show that the center frequency of the filter is 1.08 THz, the −20 dB bandwidth can reach 1.07 THz, and the average transmission rate of the stopband does not exceed 5%. Since the designed filter has a steep cut-off performance, the rising-edge or falling-edge slope cannot accurately describe the filter performance. Here we use the squared ratio that can be used to quantitatively describe the steep cut-off performance of the filter, and the formula to calculate the squared ratio is:(1)$$SF=\frac{BW(-20dB)}{BW(-3dB)}$$
where *SF* is the squared ratio and *BW* is the bandwidth. A larger squared ratio represents a better steep cut-off performance of the filter, and finally it can be calculated that the squared ratio of the designed wide-stopband filter is 0.95.

## 3. Analysis and Discussion

### 3.1. Filter Optimization

In order to further confirm the influence of filter parameters on its performance, we conducted structural optimization simulation for this design unit, and the influence of different geometric parameters on the filter performance is shown in Figure 3a. With the increasing length “L” of the metal ring, the resonant frequency of the resistive band appears red-shifted, which can be interpreted as the metal structure size becomes larger and the resonant frequency decreases. With the increasing width “w”, the starting frequency of the resistive band appears blue-shifted, while the bandwidth of the resistive band is widening, which is due to the increase of the equivalent capacitance between the resonators, while the resonant peak moves to produce a broadband. From the Figure 3b, we can see that the width of the metal ring has a very obvious effect on the performance of the stopband, so the width of the ring determines the performance of the filter. We also studied the effect of the dielectric layer thickness “d” on the filter. From the Figure 3c, we can see that as “d” increases, the bandwidth of the filter gradually becomes smaller, but the stopband performance of the filter becomes better and better, indicating that the influence of the dielectric layer thickness on the stopband performance is crucial. However, as the thickness of the dielectric layer increases, the out-of-band rejection of the filter is also affected to varying degrees. Considering the overall performance of the filter, we finally choose d = 20 μm, which not only has an ultra-wide stopband, but also has excellent absorption and out-of-band rejection.

### 3.2. Filter Performance Analysis

The goal of this paper is to construct a filter that displays broadband characteristics. We gradually stacked the designed metallic structure from one layer to four layers, as shown in Figure 4. The transmission spectra appear significantly different as the number of metal layers (n) increases. As shown in Figure 5, for a single layer (n = 1), resonance occurs at f = 0.92 THz and is a single resonance. When n = 2, it changes from a single resonance to multiple resonances, and the transmission spectrum is formed by the joint action of multiple resonance points. Similarly, for three layers (n = 3), the transmission spectrum is also formed by the joint action of multiple resonant points with resonant frequencies of 0.53, 0.66, 1.52, and 1.57 THz, respectively. It is worth noting that these apparent multi resonant spectral responses originate from the longitudinal coupling between different layers along the transmission direction, resulting in a wider stopband, better stopband performance, and steeper rising and falling edges of the stopband [18]. As shown in Figure 5, the stopband bandwidth starts to stop widening when n = 4, resulting in the characteristics of a broadband stop filter. Compared with n = 3, the filter bandwidth becomes smaller at n = 4, although the filter has better stopband performance, but the overall filter the out-of-band rejection is much less effective than at n = 3, so the final filter designed in this paper uses three layers of metal.

Due to the anisotropy of the medium, the filter is usually sensitive to the polarization angle. Whether the filter is polarization-sensitive or not is directly related to the universality of the device application and the difficulty of the operation, and this paper overcomes the polarization sensitivity problem by designing the filter structure as a centrosymmetric structure. To further investigate the transmission characteristics and polarization behavior of the wide-band filter, the transmission characteristics are simulated when the polarization angle varies from 0° to 90°. As shown in Figure 6a, the resonant frequency, bandwidth, and out-of-band rejection of the filter remain unchanged regardless of the change in polarization angle, and it can be said that the transmission characteristics hardly change, so the filter is insensitive to the polarization of incident electromagnetic waves. The response of the filter at different incident angles is also studied, as shown in Figure 6b, when the incident angle varies between 0° and 20°, the overall performance of the filter is relatively stable, and when the incident angle is larger than 20°, the overall performance of the filter is affected. It is mainly with the increase of the incident angle that an additional resonance appears in the middle of the broadband transmission spectrum, which is caused by the higher-order resonant modes in the filter.

The effective permittivity and permeability can reveal the frequency dependence of the structure, which can be calculated according to the effective medium theory [19,20]. As shown in Figure 7a,b, the effective permittivity reaches its positive maximum at about 0.53 THz, where the effective permeability is much smaller than the effective permittivity, which leads to an extreme impedance mismatch. At this point the filter generates strong electrical resonance, almost all the energy of the incident electromagnetic wave is used to sustain the oscillation of the electrons inside the structure, so that at 0.53 THz the transmission curve of the filter falls straight down, which is the beginning of the whole wide band. At about 1.12 THz, the effective magnetic permeability reaches its negative maximum, which again leads to an extreme impedance mismatch that makes it impossible for the electromagnetic wave to pass through. As shown in Figure 7c, in the 0.53 THz–1.6 THz band, the effective permittivity and effective permeability always appear in one positive and one negative state, making this band in an extreme impedance mismatch. Meanwhile, we extracted the effective impedance of the filter, as shown in the Figure 7c, in the 0.53 THz–1.6 THz band, the effective impedance is almost zero, so the vast majority of electromagnetic waves cannot penetrate the structure, thus forming a wide rejection band with high absorption.

### 3.3. Analysis of Filter Mechanism

To further analyze the origin of the wide stopband of the metamaterial filter, the electric field intensity and surface current distribution of this filter at the center frequency f = 1.08 THz are investigated in this paper. As shown in Figure 8, it is noteworthy that the order of our metal layers is arranged along the direction of the incident wave. For the first layer, it can be seen that the strong electric field is mainly distributed in the upper and lower “sidearms” of the ring, and is centrosymmetric, which has the characteristics of electric dipole resonance; while the current is mainly concentrated in the three “vertical arms” of the ring, with the same current direction. The current is mainly concentrated in the three “vertical arms” of the metal ring, and the current direction is the same. The presence of strong electrical resonance makes it possible that when the frequency of the incident electromagnetic wave is equal to the resonant frequency, almost all the energy of the incident electromagnetic wave is used to maintain the oscillation of the electrons inside the structure, so that the energy of the electromagnetic wave passing through the metamaterial filter is zero, further demonstrating that the broadband resonance arises from the electric dipole resonance [21]. For the second layer of metal, there is no longer a significant strong electric field generated; the main flow of current is the same as in the first layer of metal. However, both the electric field and the currents are much smaller compared to the first layer of metals. For the third metal layer, there is no strong electric field generated and the current distribution is the weakest among the three layers. In general, the resonance strength of the filter decreases rapidly from the top to the bottom layer. It must be noted that each metal layer plays a different role. Specifically, the first metal layer and the second metal layer of the filter work together to produce a resonant peak, while the third metal layer has a much lower surface current than the first layer, and the resonant strength is too weak to form a resonant peak. However, the third metal layer plays the role of widening the resistive band and increasing the resistive band absorption and also optimizes the steep cut-off performance of the filter.

To elucidate the advantages of the broadband filter designed in this paper over similar filters designed previously, we have conducted a comparative study of the key properties of the filter. Three aspects of unit cell size, squared ratio and in-band transmittance was compared with the filter we designed. As shown in Table 1, it can be seen that the filter designed in this paper is the smallest in terms of size compared to other filters and has the smallest in-band transmittance in the table. The squared ratio of our designed wide-stopband filter is as high as 0.95, which is the best performance of the data in the table, and 0.03 higher than the highest 0.92, which indicates that the filter designed in this paper shows better out-of-band rejection. In summary, the overall performance of the filter designed in this paper is very good. The comparison shows that the proposed structure is a suitable choice for filtering applications that require wide bandwidth and a high square ratio.

## 4. Conclusions

A wide-stopband terahertz filter is proposed based on the MDMDM structure, which exhibits good wide-stopband characteristics. The simulation results show that the filter has a center frequency of 1.08 THz, a square ratio of 0.95, and the −20 dB bandwidth reaches 1.07 THz. It is calculated that the relative bandwidth has been up to 99% and stopband absorption rate has reached more than 98%. The effect of each metal layer on the transmission performance is first discussed, thus explaining the role of each metal layer. The symmetry of the resonant structure makes the transmission characteristics of the filter insensitive to the polarization angle of the incident electromagnetic wave. By analyzing the electric field strength and surface current distribution at the center frequency of the filter, it is found that the strong electrical resonance makes the incident electromagnetic waves all used to maintain the oscillation of electrons inside the structure so that the electromagnetic wave energy through the metamaterial filter is zero. In summary, the filter has the advantages of high square ratio, ultra-wideband, and flat-top characteristics, and can be applied to sensors, terahertz communications, and electromagnetic shielding.

## Figures and Tables

**Figure 1 micromachines-13-01034-f001:**
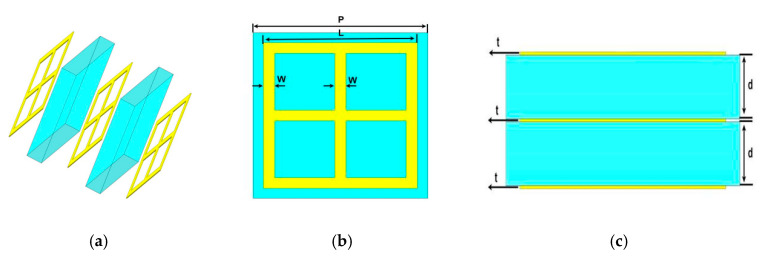
Structure of THz filter: (**a**) Layered structurer; (**b**) Top view; (**c**) Side view.

**Figure 2 micromachines-13-01034-f002:**
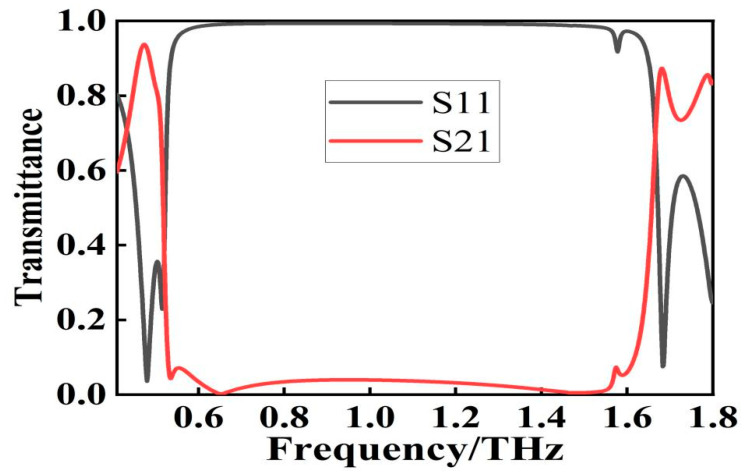
Transmittance spectrum of MDMDM structure.

**Figure 3 micromachines-13-01034-f003:**
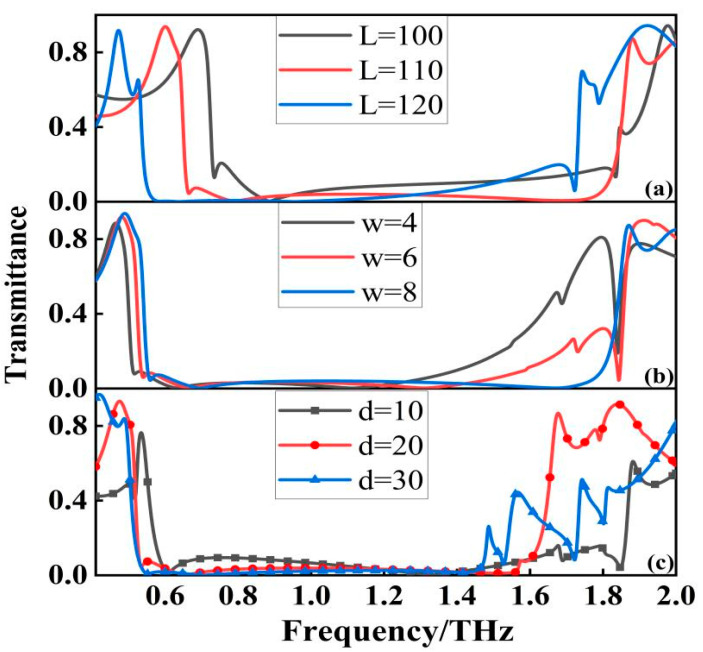
Transmittance spectra at different geometric parameters: (**a**) L = 100, L = 110 and L = 120; (**b**) w = 4, w = 6 and w = 8; (**c**) d = 10, d = 20 and d = 30.

**Figure 4 micromachines-13-01034-f004:**
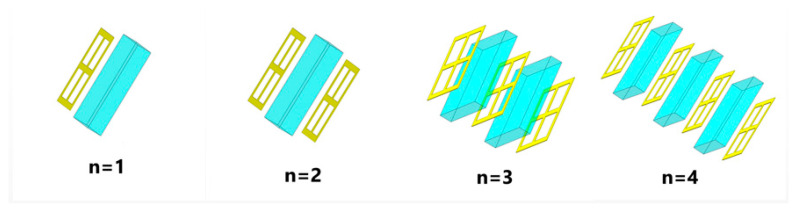
Structure with different number of metal layers.

**Figure 5 micromachines-13-01034-f005:**
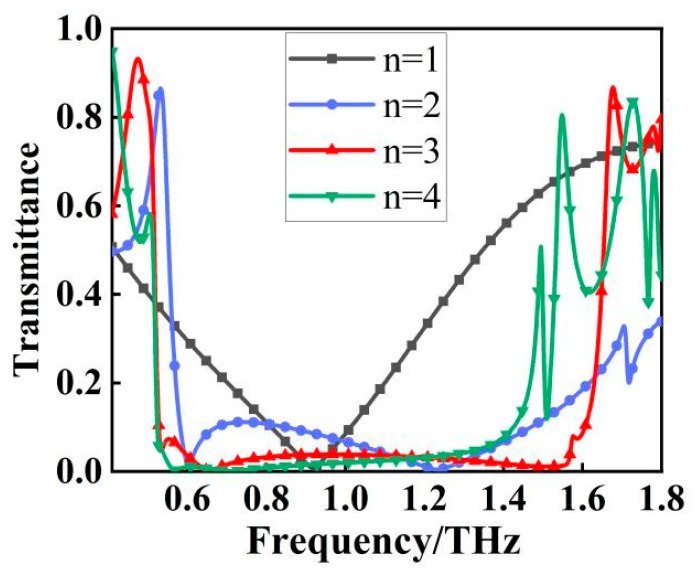
Transmittance curves corresponding to different structures.

**Figure 6 micromachines-13-01034-f006:**
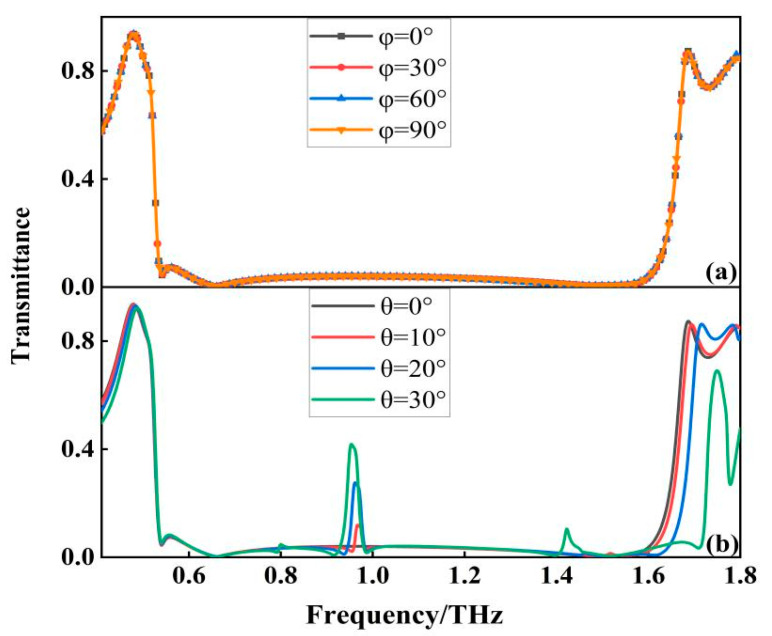
(**a**) Transmission curves corresponding to different polarization angles; and (**b**) Transmission curves corresponding to different incidence angles.

**Figure 7 micromachines-13-01034-f007:**
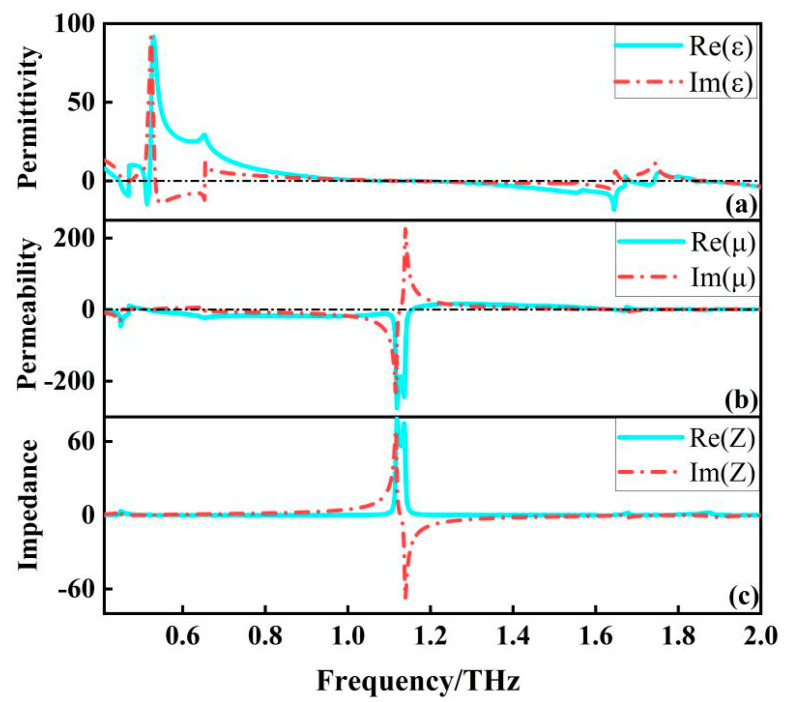
(**a**) effective dielectric constant of the filter; (**b**) effective permeability of the filter; and (**c**) effective impedance of the filter.

**Figure 8 micromachines-13-01034-f008:**
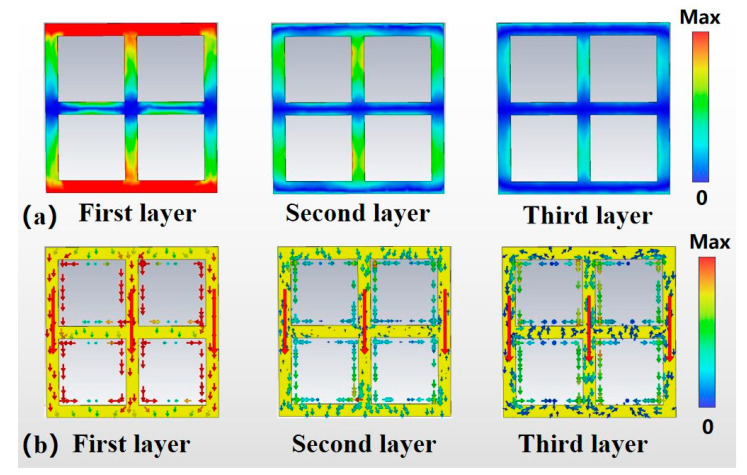
(**a**) electric field diagram at f = 1.08 THz; and (**b**) Current diagram at f = 1.08 THz.

**Table 1 micromachines-13-01034-t001:** Comparison of transmittance with other filters.

Ref.	Cell Dimension	Square Ratio	In-Band Transmittance	Relative Bandwidth
[9]	1 mm·1 mm	0.92	>5%	65%
[22]	130 μm·130 μm	0.44	>10%	43%
[23]	6.6 mm·6.6 mm	0.82	>5%	2.14%
[24]	0.4 μm·0.4 μm	0.8	>20%	40%
[25]	0.7 mm·0.7 mm	0.6	>5%	58%
[26]	140 μm·140 μm	0.56	>20%	76%
This work	125 μm·125 μm	0.95	>4%	99%

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
