# Peer review of "Design and Characterization of Wideband Terahertz Metamaterial Stop-Band Filter"

_micromachines, 2022, doi:10.3390/mi13071034_

Round 1
Reviewer 1 Report
The authors presented the design and characterization of ultra-wideband terahertz metamaterial bandstop filter.
I appreciate the work that the authors have done. However, this article missed some of the essential principles, as follows:
Title
The title is incorrect; based on the proposed frequency range, the design is not considered an ultra-wideband terahertz metamaterial!!!!!
The ultrawideband terahertz must cover the FCC licenced band, which is from 3.1 up to 10.6. We cant only say ultrawideband without having the evidence. Please check, for the paper case, I think it is "wideband"?!
Abstract
- What is MDMDM???? The Acronym should be proposed before using it??
- The authors should present the stopband absorption rate??
- The authors are required to calculate the relative bandwidth and propose it in the Abstract section?
Introduction
The introduction is insufficient and needs improvement; please discuss some metamaterial periodic structures and their ability to create the UWB stopband response for electromagnetic shielding. Frequency selective surfaces (FSSs) are considered one of them. Please see these articles, which may add value to the introduction [1,2].
[1] A miniaturised UWB FSS with Stop-band Characteristics for EM Shielding Applications. Prz. Elektrotech. 2021, 1, 142–145.
[2] A Miniaturized Quad-Stopband Frequency Selective Surface with Convoluted and Interdigitated Stripe Based on Equivalent Circuit Model Analysis. Micromachines 2021, 12, 1027. https://doi.org/10.3390/mi12091027.
Design and simulation
- Since the proposed metamaterial unit cell design structure is common (not novel), it would be nice if the authors provided the equivalent circuit model (ECM) of the proposed MTM unit cell????
- The authors are required to provide the Reflection coefficient (S11) of the proposed MTM unit cell (besides the transmittance graph).
- The authors are required to deliver the phase of transmittance?
Analysis and discussion
- The Comparison of transmittance with other filters in Table 1 is so obvious?
Please provide and compare the important parameters in terms of:
1-MTM Unit Cell dimensions.
2-Square ratio.
3-In-band transmittance.
4- Relative bandwidth.
That's all for me at this moment, and the authors are required to revise the given comments carefully. Thnaks
Author Response
请参阅附件。

Reviewer 2 Report
The paper is original but need to be improved. Fabrication of meta surfaces operating in THz is difficult but authors are advised to perform the equivalent circuit (LC) analysis to get the proof of correctness. Especially for metamaterials it is required. kindly review the papers of metasurfaces with equivalent circuit analysis.
There are English language errors in the manuscript kindly proof read the whole document to meet the standards of the MDPI.
Round 2
Reviewer 1 Report
Abstract
Line 10, please change "MDMDM (metal-dielectric-metal- dielectric-metal)" to "metal-dielectric-metal- dielectric-metal (MDMDM)".
Conclusion
Please update the conclusion with the recently added results (absorption rate and relative bandwidth).
Author Response
请参阅附件。
